# Lower Intake of Saturated Fatty Acids Is Associated with Improved Lipid Profile in a 6-Year-Old Nationally Representative Population

**DOI:** 10.3390/nu14030671

**Published:** 2022-02-05

**Authors:** Hafdis Helgadottir, Birna Thorisdottir, Ingibjorg Gunnarsdottir, Thorhallur I. Halldorsson, Gestur Palsson, Inga Thorsdottir

**Affiliations:** 1Unit for Nutrition Research, Health Science Institute, School of Health Sciences, University of Iceland, 102 Reykjavik, Iceland; hafdis24@gmail.com (H.H.); bith@hi.is (B.T.); ingigun@hi.is (I.G.); tih@hi.is (T.I.H.); 2Faculty of Sociology, Anthropology and Folkloristics, School of Social Sciences, University of Iceland, 102 Reykjavik, Iceland; 3Faculty of Food Science and Nutrition, School of Health Sciences, University of Iceland, 102 Reykjavik, Iceland; 4Children’s Hospital, Landspitali University Hospital, 101 Reykjavik, Iceland; gesturip@gmail.com

**Keywords:** blood lipids, childhood, diet quality, dietary surveys, fatty acids, nutrition

## Abstract

To strengthen the organization of new national dietary surveys and interventions in childhood, our aim was to study macronutrient intake and blood lipid profile at 6 years of age by comparing results from two earlier population-based cohorts. Subjects were *n* = 131 and *n* = 162 in the years 2001–2002 and 2011–2012, respectively. Three-day weighed food records were used to estimate diet and calculate nutrient intake. Total cholesterol, HDL-cholesterol and triacylglycerol were measured in serum and LDL-cholesterol was calculated. The average intake of saturated fatty acids (SFA) and trans FA was lower in 2011–2012 than 2001–2002 (13.3E% vs. 14.7E%, *p* < 0.001, and 0.8E% vs. 1.4E%, *p* < 0.001, respectively), replaced by a higher intake of unsaturated fatty acids. Total cholesterol and LDL-cholesterol were significantly lower in 2011–2012 than 2001–2002 (4.6 vs. 4.4 mmol/L, *p* = 0.003 and 2.8 vs. 2.5 mmol/L, *p* < 0.001, respectively). In a multiple linear regression model, one E% increase in SFA intake was related to a 0.03 mmol/L increase in LDL cholesterol (*p* = 0.04). A lower intake of saturated and trans fatty acids, replaced by unsaturated fatty acids, may have contributed to an improved lipid profile in a healthy 6-year-old population. Biological data for analysis of blood lipids are important in national dietary surveys in healthy children to monitor important health outcomes of interventions.

## 1. Introduction

A recent global review reported that the saturated fatty acid (SFA) intake of 2–7-year-old children was generally above the recommended maximum values, especially among European children [1]. While the European Food Safety Authority recommends that SFA intake should be “as low as possible”, the upper level of recommended SFA intake for children ranges from 8% of energy (E%) by FAO/WHO to 10E% by the Scientific Advisory Committee on Nutrition in the UK and the Nordic Nutrition Recommendations [2,3,4,5]. Recent literature is, however, scarce on the assessment of SFA intake in childhood and the main health outcome associated with SFA, i.e., an unfavourable blood lipid profile.

Monitoring dietary intake is of great importance for developing strategies to improve dietary habits and health. According to the American Heart Association, unhealthy diets are a major challenge for cardiovascular health promotion in children and solid fats are among the most overconsumed nutritional factors in childhood [6]. A systematic review and meta-analysis on the health effects of fatty acid intake in children concluded that reducing SFA intake between 2 and 19 years of age significantly decreased total cholesterol (TC) and low-density lipoprotein (LDL) cholesterol, with no evidence of adverse effects [7]. Total and LDL-cholesterol are established markers of the most common cause of death in Europe, i.e., cardiovascular disease (CVD) in adults [8,9] and the association between SFA and CVD is widely known [10]. However, the studies in the systematic review and meta-analysis included participants both with a wide age range and with hyperlipidaemia, so translation of the findings to healthy children should be done with a degree of caution.

National dietary surveys among children rarely include blood sample collection, limiting the ability to study how dietary intake affects blood lipids on a population level. While all age groups in childhood are important, starting school is a turning point in a child’s life and may be an important time for public health information and strategy [11].

The aim of this analysis was to compare macronutrient intake and blood lipid profile in two earlier population-based and nationally representative cohorts of healthy children, studied cross-sectionally at 6 years of age, and study possible associations between fatty acid intake and blood lipids. This research question is important for the preparation of new national dietary surveys and interventions in childhood.

## 2. Materials and Methods

### 2.1. Subjects

This study includes data from two national dietary surveys on healthy Icelandic 6-year-old children, including measurements of blood lipids. Participants in the studies were randomly selected in infancy and recruited into population-based longitudinal cohort studies on diet, growth, and health outcomes. The age of 6 years was chosen as it is the year of starting school in the country. Births were distributed over a whole-year period, residency was in all parts of the country and the participants (who represented 4–6% of live born infants in the country for the respective years) were statistically representative of the infant population of the whole country according to Statistics Iceland. The inclusion criteria in both studies were singleton birth, gestational length of 37–41 weeks, birth weight within the 10th and 90th percentiles, no birth defects or congenital long-term diseases, Icelandic parents, and mother’s participation in regular antenatal care. The methods in the two studies have previously been published in detail [12,13]. The families who had been in the cohorts in infancy until 12 months of age were invited to participate in follow-up studies when the children were 6 years old, *n* = 180 in 2001–2002 (cohort I) and *n* = 219 in 2011–2012 (cohort II) [14,15]. There was no difference between the children invited to the follow-up and the original infant cohorts when it came to the children’s mean weight and length at birth and 12 months, infant dietary intake and parents’ age, education, and body mass index (BMI). Informed written consent from the parents was obtained in infancy and at follow-up, and all individual information was processed with strict confidentiality. The studies were approved by the Icelandic Bioethics Committee, the Icelandic Data Protection Authority, and the Local Ethical Committee at Landspitali University Hospital.

### 2.2. Dietary Assessment

To assess diet and nutrient intake, all foods and fluids consumed were weighed for three consecutive days (72 h) on accurate electronic scales (PHILIPS HR 2385, Koninklijke Philips Electronics N.V, Wien, Austria). The parents or other caregivers were advised to record each food item separately and give precise information about the type of food, cooking procedure and time of serving, and to weigh and register all leftovers. All data were entered into ICEFOOD, an Icelandic calculating program designed for national dietary surveys among adults and children. Nutrient losses due to food preparation were included in the calculations. The consumption of foods and food categories were estimated in grams per day and nutrients were estimated from information about chemical content from food codes and recipes.

### 2.3. Serum Lipids

Fasting blood samples were taken from the children’s antecubital fossa. All blood samples were analysed for serum TC, high-density lipoprotein (HDL) cholesterol and triacylglycerol (TAG). TC and TAG were analysed using an enzymatic colorimetric test (Cholesterol CHOD-PAP, Roche Diagnostics, Mannheim, Germany). HDL cholesterol was measured using the same method after precipitation and centrifugation. LDL cholesterol was calculated from the serum TC, TAG and HDL concentrations expressed in mmol/L using the Friedewald formula [16], which is considered valid if TAG concentrations do not exceed 4.52 mmol/L [17]. Lipid levels were classified according to guidelines for cardiovascular health and risk reduction in children and adolescents by the US National Heart, Lung, and Blood Institute [18]. Cut-off points for acceptable and high levels, respectively, were <4.40 mmol/L and ≥5.18 mmol/L for total cholesterol, <2.85 mmol/L and ≥3.37 mmol/L for LDL cholesterol, and <0.85 mmol/L and ≥1.13 mmol/L for TAG [18].

### 2.4. Anthropometrics and Covariates

Height and weight were measured at the Children’s Hospital at Landspitali University Hospital. Subjects wore lightweight clothing and no shoes. Height was measured to the nearest 0.1 cm, using an Ulmer stadiometer, Busse design (Nersinger Straβe 18, Elchingen, Germany). Weight was measured to the nearest 0.05 kg using a Taniter BWB-620 electronic scale (2625 South Clearbrook Drive, Arlington Height, IL, USA) in cohort I and Marel Model M1100-C2 Weighing Instrument (Marel hf, Austurhraun 9 210 Gardabaer, Iceland) in cohort II. Using calculated BMI, children were classified as being normal weight or overweight/obese according to the International Obesity Task Force (IOTF) cut-off points defined to pass through a BMI of 25 at the age of 18 [19]. Cut-off points of 17.55 and 17.34 for overweight/obesity were applied for 6-year-old boys and girls, respectively.

Information on parent’s age, BMI and education were obtained from questionnaires.

### 2.5. Statistical Analysis

Descriptive analyses (mean and standard deviation, or ratios and percentages for binominal variables) were used for describing the characteristics of study participants. Independent sample t-test or the X^2^ test were used for testing for differences between continuous and dichotomous variables, respectively, in cohorts I and II. In cohorts I and II, mean differences (Δ) in fatty acid intake and 95% confidence interval (95% CI) were reported. The relationship between macronutrient intake and blood lipids was examined with multivariate linear regression. Since the directionality of the associations was the same in both cohorts, the cohorts were combined for more statistical power. In these analyses, we included as covariates energy intake (continuous), study (binary), sex (binary) and BMI (continuous) at 6 years. The macronutrients examined were total fat, SFA, monounsaturated fatty acids (MUFA), polyunsaturated fatty acids (PUFA), protein, carbohydrates, added sugar and dietary fibre. For examining the univariate association between total dietary fat intake and LDL-cholesterol in more detail and to relax the condition of linearity we used a restricted cubic-spline with two knots [20]. As these are secondary analyses, the sample size depends on the available data. A *p* value of <0.05 was considered statistically significant. The statistical analyses were performed using SAS version 9.2 and SAS Enterprise Guide (SAS Institute Inc., Cary, NC, USA).

## 3. Results

### 3.1. Characteristics of Subjects

Complete three-day food records were returned by 131 subjects (72%) in cohort I and 162 subjects (74%) in cohort II. Blood samples were collected from 137 (76%) and 145 subjects (66%), respectively. The characteristics of the subjects and their parents are presented in Table 1. In cohorts I and II, 18% and 12% of participants, respectively, were classified as overweight/obese.

### 3.2. Macronutrient Intake and Blood Lipids

As shown in Table 2, mean intake of SFA and trans fatty acids (TFA) was lower in cohort II than cohort I (*p* < 0.001), replaced by a higher intake of MUFA and PUFA. Additionally, intake of added sugar decreased and intake of fibre increased between the studies, also when taking energy into account (mean difference (95% CI) between cohorts I and II: 0.3 (0.1, 0.4) g fibre per MJ. Total cholesterol and LDL-cholesterol were lower in cohort II than cohort I (*p* = 0.003 and *p* < 0.001, respectively). There was a borderline significant trend (*p* = 0.06) towards a higher HDL-cholesterol concentration in cohort II. Further analysis revealed that the observed differences were driven by normal-weight children. Statistical power to detect significant differences between the two cohorts in the group of overweight/obese children was very low and only significant for trans fatty acids. In cohorts I and II, 17% and 13% of participants, respectively, were classified as having high TC levels, 9% and 6% of participants, respectively, were classified as having high LDL cholesterol levels, and 4% and 3% of participants, respectively, were classified as having high TG levels.

### 3.3. Relationship between Macronutrient Intake and Blood Lipids

As shown in Table 3, a one percent increase in the contribution of saturated fat to total energy intake was associated with a 0.03 mmol/L (*p* = 0.04) higher LDL-cholesterol concentration in the merged dataset of cohorts I and II, adjusted for potential confounding factors. When analysing the association separately in each cohort, the corresponding regression coefficient was 0.02 (−0.01, 0.04) in cohort I and 0.04 (0.01, 0.05) in cohort II. Similarly, the directionality of the associations observed in cohorts I and II were the same, although a slightly stronger association for SFA was observed for cohort I. Multivariate linear regression, adjusted for gender, energy intake, BMI, and study, showed that LDL levels were significantly higher when SFA intake was between 15–17E% compared to less than 11E% (β = 0.3 *p* = 0.003). The exact age of the participants was not associated with blood lipids in these children and further adjustment for age (in months) did not have any impact on the regression coefficient shown in Table 3.

When examining the association between SFA and LDL in more detail using restricted cubic spline, we observed that the increase in LDL appears to take off at around 13% SFA with the increase in LDL levelling off at around 17% SFA (Figure 1). However, only 39 out of 293 subjects had intake above 17%, so larger samples size would be needed to draw robust conclusions on whether the increase in LDL does level off at this intake.

Table 4 shows the distribution of children categorised as having acceptable, borderline–high, and high TC and LDL-cholesterol across quartiles of SFA intake. The proportion of children with acceptable/ideal cholesterol is shifted from being 59% and 74% for TC and LDL in the lowest SFA quartile, respectively, to 27% and 46% in the highest SFA quartile, respectively. This shift is primarily driven by an increased proportion of children with borderline–high TC and LDL. After adjustment for covariates, the association was no longer significant for LDL-cholesterol (*p* = 0.12), while the association remained significant for TC (*p* = 0.03).

## 4. Discussion

We hypothesise that the observed lower intake of SFA and TFA among the 6-year-olds in cohort II may have contributed to the improved lipid profile compared with cohort I studied ten years earlier. Our findings among healthy, mostly normolipidemic children, are in line with the conclusion from a systematic review and meta-analysis among children with a wide age range [7]. In the study cohorts, intake of SFA and TFA decreased by 9% and 45%, respectively, which is in line with changes in the adult population in Iceland for the same time, i.e., from 2002 to 2010–2011 [21]. While current food regulations limit the presence of TFA in the food supply, which has proven an effective CVD preventive measure for whole populations, including children [22,23], unpublished data from a new national dietary survey among Icelandic adults in 2019–2021 show a rise in SFA intake. It may be expected that children follow the same patterns. Planning a new Icelandic dietary survey in children including biomarkers of nutritional status is of utmost importance.

In a WHO systematic review and analysis, it was reported that including cis-PUFA (predominantly linoleic acid and alpha-linolenic acid) or cis-MUFA (predominantly oleic acid) in a diet as an exchange for a mixture of SFA had more favourable effects on blood lipid levels than a replacement with a mixture of carbohydrates. In particular, cis-PUFA instead of SFA decreased total and LDL cholesterol and triglycerides [24]. There are, however, different metabolic pathways of the SFAs [24,25,26], and therefore the effects vary on the blood lipid profile. In the WHO analysis, the ratios of both the total cholesterol to HDL cholesterol and the LDL cholesterol to HDL cholesterol were raised by the saturated fatty acid, lauric acid (C12:0), alone, as compared with carbohydrates [24]. Additionally, longer SFAs such as myristic acid (C14:0) and palmitic acid (C16:0) increase the blood lipids, and have unfavourable effects, as well as lauric acid (C12:0), while stearic acid (C18:0) and short SFAs of 4–10 carbons, do not [24,25,26]. Dairy is an important part of young children‘s diet and dairy foods include fatty acids, e.g., myristic acid (C14:0), with unfavourable effects on the lipid profile, but also, albeit in lower concentrations, odd chain fatty acids (OCFA) 15:0 (pentadecanoic acid) and 17:0 (heptadecanoic acid), which may have a different meaning for health [25]. Current evidence is that replacing dairy fat with polyunsaturated fat from plant-based foods has health benefits [25]. The effects of SFAs on the control of blood cholesterol levels have been suggested to be through a biological feedback mechanism. Gu and Yin (2020) describe the mechanism as SFAs diminish cholesterol intracellularly and by that give a signal about lowering cholesterol, which triggers the biosynthesis of cholesterol [26]. Therefore, when cholesterol lowers in cells, the plasma LDL-cholesterol is raised. The step of lowering intracellular cholesterol is by a suppression of LDL endocytosis mediated by LDL receptors and retarding cholesterol transport from plasma membrane to the endoplasmic reticulum. This system effectively regulate the synthesis by sensing cholesterol fluctuations in cells by activating sterol regulatory element-binding protein 2 pathway and degrading 3-hydroxy-3-methyl-glutaryl coenzyme A reductase [26].

It is not certain to what extent a reduction in cholesterol in children will decrease later risk of coronary heart disease, but significant tracking from childhood to adulthood has been reported in many studies [18]. For example, in the Bogalusa Heart Study, children with LDL cholesterol above 3.35 mmol/L (corresponding to our classification as high LDL levels) had significantly higher prevalence of adult dyslipidaemia compared with children with LDL cholesterol below 2.84 mmol/L [27]. In our cohorts, 9% and 6% of children in cohort I and II, respectively, were classified as having high LDL levels. It has been estimated that a reduction of one mmol/L in cholesterol will reduce CHD mortality rates by approximately 50% in middle-aged individuals [8]. According to a study of the Icelandic Heart Association, cholesterol concentration decreased from 6.01 mmol/L to 5.14 mmol/L in the adult population from 1981 to 2006 [28]. At the same time, the contribution of saturated fatty acids plus trans fatty acids to total energy intake of adults reduced from 19.0E% to 15.2E% [21,29]. CHD mortality rates did also decline considerably in Iceland during that time [28], by 80% in men and women. The decline was mostly attributed to risk factor reductions, i.e., cholesterol, smoking and physical inactivity. The 0.3 mmol/L reduction in average LDL cholesterol concentration in 6-year-olds seen in the present study on the population level may be of clinical relevance, as it will have decreased the number of children considered to be at risk of adult dyslipidemia. Even though recent scientific publications and recommendations confirm the former knowledge and advice on lowering SFA and TFA to improve lipid profile in childhood [1,7] and adults [30], there are still doubts about the influences of exchanging SFA for PUFA [31]. Further, there are different ways to present SFA in a diet, and an increase in the percentage intake of SFA may improve the lipid profile provided the diet is a weight loss diet focusing on lower energy intake with lower intake of refined carbohydrates [32].

Childhood is further a critical period of establishing dietary habits that tend to track until later in life, when they may affect CVD risk [33,34]. Therefore, both continuous monitoring of dietary habits and establishment of healthy dietary habits in infancy and childhood are vital [35,36]. Further, addressing food and nutrition literacy and adapting nutritional advice to local intake patterns rather than specific macronutrient intake may improve children’s adherence to dietary guidelines [37,38]. In children, as in adults, blood lipids are considered to be regulated by an interplay of genetic, dietary and lifestyle factors [39]. Weight regulation and increased physical activity may also be beneficial for blood lipids [40]. For the age group studied, i.e., 6-year-olds starting school, we believe it may be a good time for teaching about food and nutrition and health promotion.

There are several limitations associated with the interpretation of the results from two independent cohort studies conducted 10 years apart. However, the present study brings important messages to health authorities implementing different strategies in order to improve health at the population level. The results can therefore be considered of value when developing and supporting concepts in public health nutrition related to improved fat quality. Given the relatively low number of subjects in the present analysis, the confidence intervals in our estimation of the association between one E% increase in the intake of saturated fat and cholesterol intake is broad. However, our results are in line with the results presented by large cohort studies and intervention studies as well as the recent meta-analysis on a wider age range of participants.

In conclusion, this study showed improved dietary fat quality and blood lipid profile among Icelandic 6-year-old children in a nationally representative cohort studied in 2010–2011 (cohort II) as compared to 2001–2002 (cohort I). Lower intake of saturated fat and trans fatty acids, replaced by unsaturated fat, might have contributed to improved lipid profile at the population level. This shows the relevance of including health-related biomarkers in national dietary surveys in childhood for following and better understanding trends in public health and to be better able to give scientifically based advice and prepare interventions in childhood.

## Figures and Tables

**Figure 1 nutrients-14-00671-f001:**
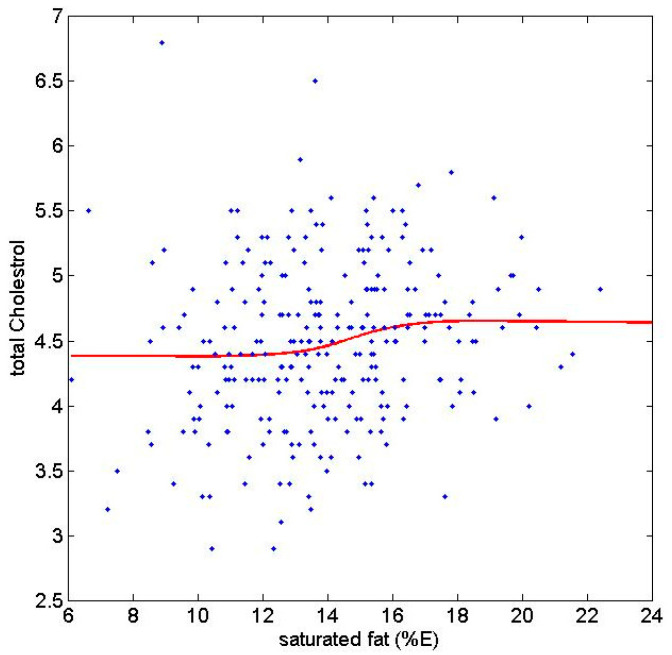
The association between saturated fatty acids and LDL-cholesterol examined by restricted cubic spline for both cohorts of 6-year-old children combined.

**Table 1 nutrients-14-00671-t001:** Characteristics of the 6-year-old participants and their parents in two cohorts conducted 10 years apart.

Variables	Cohort I (*n* = 137) Mean (SD)	Cohort II (*n* = 145) Mean (SD)	Mean Difference Δ (95% CI)
Child			
Age (months)	72.3 (1.6)	73.4 (3.2)	1.1 (0.5, 1.7) *
Weight (kg)	23.0 (3.4)	23.0 (3.7)	−0.1 (−0.9, 0.8)
Height (cm)	119.0 (4.4)	120.0 (4.9)	1.0 (−0.0, 2.1)
BMI (kg/m^2^)	16.1 (2.3)	15.9 (1.8)	−0.2 (−0.6, 0.3)
Parents			
Mother’s age (years)	35.7 (5.4)	36.3 (4.9)	0.6 (−0.7, 2.0)
Father’s age (years)	37.7 (5.9)	38.9 (5.9)	1.2 (−0.4, 2.8)
Mother’s BMI (kg/m^2^)	25.5 (4.4)	24.8 (4.9)	−0.6 (−2.0, 0.7)
Father’s BMI (kg/m^2^)	26.5 (3.2)	26.2 (3.2)	−0.3 (−1.4, 0.8)
Mother’s education ≥ 12 years, ^1^ *n* (%)	70 (74)	121 (81)	0.7 (0.2, 1.3)
Father’s education ≥ 12 years, ^1^ *n* (%)	76 (80)	111 (76)	1.4 (0.8, 2.5)

* *p* < 0.05. ^1^ Presented as number and percentages. Δ = mean difference; BMI = body mass index; CI = confidence interval; SD = standard deviation.

**Table 2 nutrients-14-00671-t002:** Intakes and serum lipid concentrations of the 6-year-old participants in the two cohorts conducted 10 years apart.

Variables	Cohort I Mean (SD)	Cohort II Mean (SD)	Mean Difference Δ (95% CI)
Intake of energy and energy-giving nutrients	*n* = 131	*n* = 165	
Energy (kcal)	1494 (308)	1543 (324)	49 (−25, 122)
Total fat (E%)	33.1 (5.4)	32.2 (4.9)	−0.9 (−2.1, 0.3)
SFA (E%)	14.7 (3.1)	13.3 (2.7)	−1.3 (−2.0, −0.7) *
TFA (E%)	1.4 (0.5)	0.8 (0.3)	−0.7 (−0.7, −0.6) *
MUFA (E%)	9.5 (1.7)	10.1 (1.8)	0.6 (0.2, 1.0) *
PUFA (E%)	3.8 (1.2)	4.7 (1.5)	0.9 (0.6, 1.2) *
Omega-3 PUFA (E%)	0.9 (0.4)	1.2 (0.6)	0.3 (0.2, 0.4) *
Omega-6 PUFA (E%)	2.9 (0.9)	3.4 (1.2)	0.5 (0.3, 0.8) *
Omega-6/Omega-3 PUFA ratio (%)	3.6 (1.3)	3.2 (1.2)	−0.4 (−0.6, −0.1) *
Protein (E%)	15.6 (2.8)	15.4 (2.9)	−0.3 (−0.9, 0.4)
Carbohydrates (E%)	50.8 (5.3)	50.3 (5.5)	−0.5 (−1.8, 0.7)
Added sugar (E%)	12.7 (4.3)	11.2 (4.5)	−1.6 (−2.6, −0.5) *
Fibre (g)	11.1 (3.2)	13.2 (4.0)	2.1 (1.3, 3.0) *
Serum lipid concentration	*n* = 137	*n* = 145	
Total cholesterol (mmol/L)	4.6 (0.6)	4.4 (0.6)	−0.2 (−0.4, −0.1) *
LDL-cholesterol (mmol/L)	2.8 (0.5)	2.5 (0.6)	−0.3 (−0.4, −0.1) *
HDL-cholesterol (mmol/L)	1.5 (0.3)	1.6 (0.3)	0.1 (0.0, 0.2)
TAG (mmol/L)	0.6 (0.2)	0.6 (0.2)	0.02 (0.0, 0.1)

* *p* < 0.05. Δ = mean difference; CI = confidence interval; HDL = high density lipoprotein; LDL = low density lipoprotein; MUFA = monounsaturated fatty acids; PUFA = polyunsaturated fatty acids, SD = standard deviation; SFA = saturated fatty acids; TAG = triacylglyceride; TFA = trans fatty acids.

**Table 3 nutrients-14-00671-t003:** The relationship between macronutrient intake and total and LDL-cholesterol assessed by linear regression analysis for both cohorts of 6-year-old children combined.

	Total Cholesterol (mmol/L)				LDL-Cholesterol (mmol/L)			
	Unadjusted		Adjusted ^1^		Unadjusted		Adjusted ^1^	
	β	95% CI	β	95% CI	β	95% CI	β	95% CI
Total Fat (E%)	0.02	0.006, 0.03 *	0.02	0.004, 0.03 *	0.02	0.004, 0.03 *	0.02	0.003, 0.03 *
SFA (E%)	0.04	0.01, 0.06 *	0.03	0.002, 0,05 *	0.03	0.01, 0.06 *	0.03	0.002, 0.05 *
MUFA (E%)	0.01	−0.03, 0.06	0.02	−0.02, 0.07	0.01	−0.02, 0.05	0.03	−0.01, 0.07
PUFA (E%)	0.02	−0.03, 0.07	0.04	−0.01, 0.09	0.00	−0.05, 0.05	0.03	−0.02, 0.07
Protein (E%)	0.006	−0.02, 0.03	0.01	−0.02, 0.04	0.01	−0.01, 0.03	0.01	−0.01, 0.04
Carbohydrates (E%)	−0.02	−0.03, −0.003 *	−0.02	−0.03, −0.01 *	−0.01	−0.03, −0.002 *	−0.02	−0.03, −0.01 *
Added sugar (E%)	−0.003	−0.02, 0.01	−0.01	−0.03, 0.01	−0.01	−0.02, 0.01	−0.02	−0.03, 0.001
Fibre (g)	−0.01	−0.03, 0.01	−0.01	−0.04, 0.01	−0.01	−0.03, 0.01	−0.01	−0.03, 0.02

* *p* < 0.05. ^1^ Adjusted for sex, energy intake, BMI, and cohort (I or II). β = beta coefficients; CI = confidence interval; HDL = high density lipoprotein; LDL = low density lipoprotein; MUFA = monounsaturated fatty acids; PUFA = polyunsaturated fatty acids, SD = standard deviation; SFA = saturated fatty acids; TAG = triacylglyceride; TFA = trans fatty acids.

**Table 4 nutrients-14-00671-t004:** Distribution of children according to classification of lipid levels across quartiles of saturated fatty acid intake for both cohorts of 6-year-old children combined.

	Total Cholesterol (mmol/L)				LDL-Cholesterol (mmol/L)			
SFA (E%)	Acceptable	Borderline	High	*p*-Value	Acceptable	Borderline	High	*p*-Value
Q1 (<12E%)	59%	30%	11%		74%	18%	8%	
Q2 (12–14E%)	49%	35%	15%		62%	31%	8%	
Q3 (14–16E%)	51%	34%	15%		60%	41%	9%	
Q4 (>16E%)	27%	55%	18%		46%	48%	6%	
unadjusted ^1^				0.02				0.03
adjusted ^2^				0.03				0.12

^1^ Chi square test. ^2^ Adjusted for sex, energy intake, BMI, and cohort (I or II) using logistic regression analysis.

## Data Availability

The data presented in this study are available on request from the corresponding author. The data are not publicly available due to the nature of the original ethical approvals.

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
