# Peer review of "Lower Intake of Saturated Fatty Acids Is Associated with Improved Lipid Profile in a 6-Year-Old Nationally Representative Population"

_nutrients, 2022, doi:10.3390/nu14030671_

Round 1
Reviewer 1 Report
Authors investigated the effect of food intake on lipid profile in a 6-year-old children. Interesting results were obtained pointing out the potential importance of the results to direct further strategies for improve the health of this population. Please, consider the following criticisms and suggestions:
- a) Introduction. Why did the authors choose children at 6 years old? Why not a temporal analysis of this population? Are other ages important? What are the limittations of studying only children at 6 years old? A clear justificative and rationality have to be provided. In addition, a background about the importance of food intake on health promotion has to be provided.
- b) Materials and methods. Authors have to briefly describe the inclusion criteria, even it was already published in previous studies. This information is fundamental to understand the study. For example, did the children present any physiological disturbances, including overweight, obesity, or comorbidity? Any disease (metabolic, inflammatory, or chronic disease)? Etc. These factors can greatly interfer with the results.
- c) Statistical analysis. 1) Please, inform how the sample size was determined. 2) Describe what the tests were used to detect statistical differences and significance levels.
- d) Results. 1) In the Tables 1-3, indicate what parameters are different. 2) It will be interesting to analyse subgroups in the cohort 1 and 2, including chidren’s BMI, parents’s BMI, and parents’s education. These analyses will be important to show the interference of these parameters on the results.
- e) Discussion. Discussion is limited to the results of the study. It has to be improved according to the suggestions and criticisms above. Any discussion about the children’s or parents’s characteristis is presented. Mechanistic pathways are also lacking trying to explain the results.
Reviewer 2 Report
1.Association between saturated fatty acids and lipid profile has been studied in previous researches, as well as in children and adolescents. Thus the research question in this paper is less innovative.
2.Data in this study were two cross-sectional surveys from two different cohorts, and there may be large differences in characteristics of the participants involved, study design and sampling methods, which are incomparable and could not account for the diet and lipid changes from 2002 to 2010-2011 in Iceland. In addition, the sample sizes are too small to represent the national level.
3.Author is suggested to provide the data support in supplementary file to prove that the trend was similar in both two studies and could be combined.
4.Linked research in this study is suggested to adjust for more comprehensive confounding factors, including age, region and factors associated with lipid profile (such as sedentary time, physical activity).
5.Elevated LDL levels in this paper do not mean dyslipidemia. As a result, author is suggested to divide the outcome into categorical variable in order to distinct pathological and physiological states.
6.Discussion section is advised to add comparison with previous studies and an in-depth interpretation of the underlying mechanism. The references are outdated and need to be updated.
Round 2
Reviewer 1 Report
Authors have addressed most suggestions and criticisms in the new version of the manuscript. I have no additional comments.
Author Response
Thank you.
Reviewer 2 Report
- In response to the 5th comment, although author has supplemented the classification description of the outcome variable, what I mean is that the author is suggested to use the categorical outcome variable as outcome in the association analysis (3.3 Relationship between macronutrient intake and blood lipids). To be honest, the content of this paper is not deep enough; thus adding the content above is helpful to rich the paper.
- The author did not update the references, and the obsolete references accounted for about the half in both the revision and the original papers.
